# Preparation and Study of Sulfonated Co-Polynaphthoyleneimide Proton-Exchange Membrane for a H2/Air Fuel Cell

**DOI:** 10.3390/ma13225297

**Published:** 2020-11-23

**Authors:** Ulyana M. Zavorotnaya, Igor I. Ponomarev, Yulia A. Volkova, Alexander D. Modestov, Vladimir N. Andreev, Alexei F. Privalov, Michael Vogel, Vitaly V. Sinitsyn

**Affiliations:** 1A.M. Prokhorov Institute of General Physics RAS, Vavilova st. 38, 119991 Moscow, Russia; ulyanamzav@mail.ru; 2A.N. Nesmeyanov Institute of Organoelement Compounds, Vavilova st. 28, GSP-1, 119991 Moscow, Russia; gagapon@ineos.ac.ru (I.I.P.); yvolk@ineos.ac.ru (Y.A.V.); 3A.N. Frumkin Institute of Physical Chemistry and Electrochemistry, Leninsky pr. 31, 119071 Moscow, Russia; amodestov@mail.ru (A.D.M.); vandr@phyche.ac.ru (V.N.A.); 4Institute of Condensed Matter Physics, Technische Universität Darmstadt, Hochschulstr. 6, 64289 Darmstadt, Germany; alexei.privalov@physik.tu-darmstadt.de (A.F.P.); michael.vogel@physik.tu-darmstadt.de (M.V.); 5Inenergy LLC, Electrodnaya str., 12-1, 111524 Moscow, Russia

**Keywords:** sulfonated polynaphthoyleneimides membrane, proton conductivity, water self-diffusion coefficient, NMR diffusometry, membrane–electrode assembly, current–voltage characteristic, MEA power output

## Abstract

The sulfonated polynaphthoyleneimide polymer (co-PNIS70/30) was prepared by copolymerization of 4,4′-diaminodiphenyl ether-2,2′-disulfonic acid (ODAS) and 4,4’-methylenebisanthranilic acid (MDAC) with ODAS/MDAC molar ratio 0.7/0.3. High molecular weight co-PNIS_70/30_ polymers were synthesized either in phenol or in DMSO by catalytic polyheterocyclization in the presence of benzoic acid and triethylamine. The titration reveals the ion-exchange capacity of the polymer equal to 2.13 meq/g. The membrane films were prepared by casting polymer solution. Conductivities of the polymer films were determined using both in- and through-plane geometries and reached ~96 and ~60 mS/cm, respectively. The anisotropy of the conductivity is ascribed to high hydration of the surface layer compared to the bulk. SFG NMR diffusometry shows that, in the temperature range from 213 to 353 K, the ^1^H self-diffusion coefficient of the co-PNIS_70/30_ membrane is about one third of the diffusion coefficient of Nafion^®^ at the same humidity. However, temperature dependences of proton conductivities of Nafion^®^ and of co-PNIS_70/30_ membranes are nearly identical. Membrane–electrode assemblies (MEAs) based on co-PNIS_70/30_ were fabricated by different procedures. The optimal MEAs with co-PNIS_70/30_ membranes are characterized by maximum output power of ~370 mW/cm^2^ at 80 °C. It allows considering sulfonated co-PNIS_70/30_ polynaphthoyleneimides membrane attractive for practical applications.

## 1. Introduction

Fuel cells with proton-exchange membranes (PEMFC) are the most commercially successful instrument for direct generation of electricity from a hydrogen–oxygen chemical system, excluding combustion [1,2,3,4,5]. To generate electricity directly from a chemical reaction, the fuel (hydrogen) is supplied to the anode compartment of the PEMFC, and an oxidizer (air or oxygen) is supplied to the cathode compartment of the cell. An essential part of the PEMFC is a Membrane Electrode Assembly (MEA). It consists of two gas diffusion electrodes (GDE) attached to the proton-exchange membrane (PEM). The GDEs contain catalyst layers in which triple phase boundaries-electronic conductor/ionic conductor/gas phase are arranged [6,7,8]. In the anode catalyst layer, hydrogen is oxidized to give protons with the release of electrons in the outer circuit. Protons cross the membrane towards cathode where they are consumed in electrochemical reaction of oxygen reduction to water by electrons which arrive from the external circuit. Due to potential difference between anode and cathode GDEs the electrons produce useful work on the load in the external circuit.

The PEM is the most important element of the MEA, which determines its performance, lifetime and external operating conditions. It should be gas-tight in order to separate the gas flows of the fuel and oxidizer and allow only proton transport through the membrane (electronic conductivity of PEM is negligible compared to proton conductivity). Perfluorinated sulfopolymers of the Nafion^®^ type, developed by DuPont in the 1960s, are the most widely used polymer membranes in PEMFC [9]. However, their complexity and high manufacturing costs, which result from the presence of fluorine in their chemical structure, significantly limit the commercialization and wide distribution of PEMFC so that only about 10 companies are currently engaged in the production of such membranes. As a possible alternative, hydrocarbonate (polyarylene) polymers are proposed. These systems are attractive for the use as PEM because their chemical structure is fluorine free and the fabrication costs are much lower. A lot of effort has been undertaken to develop PEMs derived from sulfonated polynaphthoyleneimides (SPNI), especially based on industrial 1,4,5,8-naphthalene tetracarboxylic acid dianhydride (NTDA) with six-membered imide rings, which have been reported to be promising candidates for PEMFC [10,11,12,13]. Mercier et al. developed sulfonated block copolynaphthoylene imides (SPNIs) from 1,4,5,8-naphthalene tetracarboxylic dianhydride (NTDA), 2,2′-benzidine disulfonic acid (BDSA) and conventional non-sulfonated diamines. These membranes showed reasonable performance in a H_2_/O_2_ fuel cell system at 60 °C for over 3000 h. However, their proton conductivity was significantly lower due to their low ion exchange capacity (IEC = 1.30 meq/g) and low hydrolytic stability (which was important for maintaining membrane durability towards water in acidic environments). The most stable co-SPNIs membranes from NTDA are sulfonated in a special way where triazole-containing diamines (IEC = 2.33 meq/g) are incorporated in their chemical structure. Such MEA was found fairly stable over 5000 h operation at 80 °C and 0.2 A/cm^2^ load. The open circuit voltage (OCV) of the cell under testing did not change significantly in the course of 5000 h, indicating sufficient hydrolytic stability and low gas crossover through the membrane [4]. The main problem concerning the use of co-SPNIs and classical PNI polymers in fuel cell applications is that all polyimides are known to be sensitive to hydrolysis [10,14,15,16]. The introduction of sulfonate groups along the polymer chains increases the overall hydrophilicity and, consequently, the rate of water diffusion within the structure, which contributes to the hydrolytic process. Moreover, many linear high molecular weight SPNIs are water soluble. Random and block co-SPNIs membranes characterized by IEC ≥ 2.0 meq/g dissolve or fragment in water at elevated temperatures (>50 °C) [17]. The mechanism of degradation of SPNIs in water and the stages of their equilibrium were studied in detail in [14].

Taking into account that the formation of 5- or 6-membered imide rings are mainly catalyzed by carboxylic acids [18], we decided to synthesize co-SPNIs based on NTDA, 4,4′-diaminodiphenyl ether-2,2′-disulfonic acid (ODAS) and 4,4’-methylenebisanthranilic acid (MDAC) as non-sulfonated monomer. MDAC is an important component that plays the role of internal catalyst, which rebuilds the polymer chain in the case of its degradation by hydrolysis. The NTDA-ODAS polymer membrane (IEC = 3.37 meq/g) has a high proton conductivity (σ~0.1 S cm^−1^ in water) even at room temperature, but it does not retain its integrity and dissolves in water after one day [17]. The NTDA-MDAC polymer is insoluble in water and contains a carboxyl group, which plays the role as a catalyst during PNIs synthesis and at equilibrium stages during imide cycle hydrolysis [14,17,18]. Such combination of monomers can improve the overall hydrolytic stability of co-SPNIs. Therefore, various high molecular weight NTDA/ODAS/MDAC co-SPNIs have been synthesized with ODAS/MDAC molar ratios of 0.5–0.9/0.1–0.5. All polymers have shown good film-forming properties and could be tested as proton-conducting material for H_2_/air PEMFCs.

In this work, we prepare membranes and MEAs using the copolymer with the ODAS/MDAC molar ratio of 0.7/0.3 = 70/30, which we denote as co-PNIS_70/30_ hereinafter. The synthesis was carried out in phenol and in dimethyl sulfoxide (DMSO) by catalytic polyheterocyclization in the presence of benzoic acid and triethylamine to obtain high molecular weight polymers, characterized by viscosity (η) in sulfuric acid in the range of 1.0–3.0 dL/g according to the following reaction Scheme 1 [19,20]:

The goal of our study was to find an efficient method for MEA fabrication using co-PNIS polymers. Accordingly, we optimized the fabrication of MEA (based on co-PNIS_70/30_) and studied their current–voltage characteristics, diffusion and conductivity in-plane and through-plane of the polymer film.

## 2. Materials and Methods

### 2.1. Co-PNIS with ODAS/MDAC = 70/30 Synthesis

DMSO, phenol, triethylamine, 4,4′-diaminodiphenyl ether, benzoic acid, sulfuric acid, oleum (65% SO_3_) and sodium hydroxide were purchased from Acrus (Moscow, Russia). 6,6-bis-methylenedianthranylic acid (MDAC) (Vitas-M Laboratory) and 1,4,5,8-napthalenetetracarboxylic acid dianhydride (NTDA) (Sigma Aldrich, St. Louis, MO, USA) were used without further purification. 4,4’-Diaminodiphenyl oxide-2,2′-disulfoacid (ODAS) was synthesized as described in [17].

For the preparation of co-SPNIs polymer films, the chemical components ODAS (0.5044 g, 0.0014 M), MDAC (0.1718 g, 0.0006 M), NTDA (0.5364 g, 0.002 M), benzoic acid catalyst (0.34 g, 0.0028 M), triethylamine solubilizer of ODAS (0.303 g, ~0.003 M) and phenol or DMSO as solvents (10.0 g) were placed in a three-piece flask equipped with a stirrer and a capillary for introducing argon [19,20]. The reaction mass mixture was heated in an argon flow to 80 °C while stirring until all monomers were dissolved. Then, the temperature was increased, and the reacting mixture was kept for 24 h at 120–140 °C. Afterwards, it was cooled to 100 °C and diluted with phenol or DMSO to 5% by weight concentration. To cast the membranes, the reaction solutions were filtered and then poured onto a glass substrate for drying at 60 °C. The films were removed from the glass and additionally kept for 2 h in vacuum at 150 °C in order to eliminate residual solvent.

The concentration of the sulfo-groups (SO^3−^) was calculated from the known concentration of the chemical components in the synthesis and verified by titration of a polymer film. Samples for titration were pre-weighed on an analytical balance (BEL Engineering, Monza, Italy), and then placed in a glass with distilled water and kept for 1 h at (80 ± 3) °C. After that, the samples were transferred into flasks with 20 mL of 1 M NaCl (LLC "Chemistry XXI century", Moscow, Russia) solution and kept for 30 min under normal conditions. Then, 2 drops of an indicator (phenolsulfophthalein C_19_H_14_O_5_S) were added to each flask and titrated with 0.01 M NaOH solution.

The IEC was calculated using the equation:(1)$$IEC=\frac{0.001\times V_{NaOH}\times C_{NaOH}}{m}$$
where *V_NaOH_* is the volume of the alkali solution, mL; *C_NaOH_* is the concentration of the alkali solution, mol/L; and *m* is the dry membrane weight, g. We found that the IEC of the co-PNIS_70/30_ membrane amounts to 2.13 meq/g.

### 2.2. Conductivity Measurements

The impedance spectroscopy method was applied to determine the proton conductivity. The measurements were carried out on an SP-240 Biologic spectrometer (Bio-Logic, Seyssinet-Pariset, France) in the frequency range from 1 Hz to 500 kHz using an amplitude of 50 mV at room temperature in air with 100% humidity. To check the possible anisotropy of the conductivity which can result from membrane preparation and further its hydration, we measured the impedance spectra for in- and through-plane geometries of the polymer film. The measuring cells are shown in Figure 1a,b.

For in-plane geometry (Figure 1a), the Teflon base with slots for humidified gas circulation was used, which allows maintaining at fixed sample humidity. Platinum wires acting as electrodes (1) are located in the grooves between the slots. To improve the reversibility of the membrane–electrode interface for charge transfer, the platinum wire electrodes (5 and 6) were coated with the Pt/C catalyst used in the GDEs (see Section 2.4). A round hole in the Teflon plates was made for clamping screws. The four-point scheme was used. During the measurements, the cell was placed in a closed container with 100% humidity. The specific value of proton conductivity in this geometry is:(2)$$\sigma =\frac{1}{R}\frac{l}{d\times h}$$
where *l* = 10 mm is the distance between the potential electrodes (6) in Figure 1; *h* = 8 mm is the width of the membrane; *d* is its thickness; and *R* is the resistance of the sample, corresponding to the high-frequency cutoff of the impedance on the real *Z*’ axis.

Figure 1b demonstrates a cell sketch for the through-plane impedance spectra measurements of polymer films. In this case, the measurements were carried out by means of a two-contact scheme. The holes in the platinum foil (3) were made for humid air access to the membrane. The specific conductivity in this geometry was calculated as:(3)$$\sigma =\frac{1}{R}\frac{d}{S}$$
where *d* is the thickness of the membrane; *S =* 0.25 cm^2^ is the area of the Pt foil (2); and *R* is an extrapolation to the real *Z*’-axis of an impedance spectrum corresponding to the volume response of the film.

Samples of different thickness were initially investigated at room temperature and 100% humidity to check the accuracy of the measurements. The studies showed that the high-frequency cutoff in the impedance spectra changes near proportionally to the film thickness for through-plane geometry (Table 1) and reversely proportional to the thickness for in-plane geometry (Table 1). It indicates the correctness of the measurements and the determination of the bulk conductivity of polymer films for both geometries.

### 2.3. Diffusion Measurements

To quantify the effect of the temperature on the dynamic properties, the self-diffusion coefficients of the protons were measured using ^1^H Static Field Gradient (SFG) NMR diffusometry [21,22]. SFG NMR exploits the fact that the resonance frequencies of the observed spins change as a result of displacements with respect to the magnetic field gradient. This leads to a decrease in the echo-signal amplitudes, depending on the diffusion time. SFG NMR, as compared to the pulsed field gradient analog, allows us to use significantly higher field gradients and, thus, to measure slower diffusion and to reach lower temperatures. For our measurements, we mainly used the stimulated-echo pulse sequence (see Figure 2). It has three pulses that separate a mixing time *t_m_* from the two enclosing evolution times *t_e_* and produce a stimulated echo.

The echo amplitude *S* decreases due to free diffusion and spin relaxation in accordance with:(4)$$S\left(t_{e},t_{m},g\right)=S_{0}\cdot \exp\left(\frac{-t_{m}}{T_{1}}\right)\cdot \exp\left(\frac{-2t_{e}}{T_{2}}\right)\cdot \exp\left[{\left(\gamma gt_{e}\right)}^{2}\cdot \left(\frac{2}{3}t_{e}+t_{m}\right)\cdot D_{NMR}\right],$$ Here, *γ* is the gyromagnetic ratio of the ^1^H nuclei, *g* is the magnetic field gradient and *D_NMR_* is the self-diffusion coefficient. Moreover, *T*_1_ and *T*_2_ denote the spin–lattice and spin–spin relaxation times, respectively, which limit the diffusion time and, thus, the dynamic range of the experiment. In favorable cases, self-diffusion coefficients down to D = 10^−15^ m^2^/s can be measured [21,22].

To eliminate echo damping due to spin relaxation or other unwanted effects such as polarization transfer between mobile and immobile spins, we divide the results of identical measurements performed for the same resonance frequency of 60 MHz but different magnetic field gradients, *G* = 178 T/m and *g =* 41 T/m. Because the unwanted damping factors are independent of the field gradient, they are cancelled out by this division. Thus, the divided data decay exclusively due to proton diffusion in an effective magnetic field gradient (G^2^ − *g*^2^):(5)$$\frac{S\left(t_{m},\tau ,G\right)}{S\left(t_{m},\tau ,g\right)}=S_{0}\cdot \exp\left[-{\left(\gamma \tau \right)}^{2}\left(G_{}^{2}-g_{}^{2}\right)\cdot \left(\frac{2}{3}\tau +t_{m}\right)D_{NMR}\right],$$

Such measurements can be performed using a special setup with an anti-Helmholtz arrangement of superconducting coils. It has four positions with identical Larmor frequency but two different gradients, which are available by correctly adjusting the position of the probe head along the magnet axis. The length of the 90^°^ pulses was 0.7 μs. The measurements were carried out in the temperature range from 213 to 353 K. The temperature was set with an accuracy of ±1 K and stabilized to ±0.1 K using a N_2_ flow cryostat. Figure 3 shows typical experiment results for the original and divided data together with a fit using Equation (5).

### 2.4. MEAs Fabrications and Investigations of Their Voltage-Current Characteristics

MEAs were fabricated by attaching GDEs to the membrane. In some cases, attachment of the GDEs was achieved during the cell assembly (MEA with attaching electrodes). In other cases, GDEs were hot-pressed to the membrane prior to cell assembly (MEA with pressing electrodes). A Pt/C catalyst with a Pt content of 48.98% (manufactured by InEnergy LCC, Moscow, Russia) was used [23,24]. The catalyst layer on the surface of the GDE was prepared by spraying catalyst ink on the gas diffusion layer Freudenberg I2C8 (Freudenberg FCCT SE & Co KG, Weinheim, Germany). The active area of the tested MEAs was 1.21 cm^2^. The electrochemical investigations of fuel cells were carried out on the symmetrical MEAs with a platinum loading of 0.4 mg/cm^2^ at both the anode and cathode GDEs. The parameters of fabrication of MEAs with pressing GDEs were optimized during separate studies. The best performance was obtained under the pressing conditions: P = 1050 N/sm^2^, T = 80 °C, t = 180 c. These conditions of MEA fabrication were taken as the optimal parameters.

The MEAs electrochemical characteristics were measured using the Arbin test facility at a hydrogen and air flow rate of 200 mL/min for each gas. The open-circuit potential for all MEA tests was ~1 V. Initially, the MEA was subjected to a pulse activation (break-in) in accordance with the procedure described in [25]. After 10–12 pulse activation cycles, the MEA was potentiostatically loaded at *U*_load_ = 0.5 V and the time-dependent current *I*(t) was measured. Figure 4 shows an exemplary *I*(t) curve in the potentiostatic break-in mode for a MEA with pressing electrodes together with the corresponding output power *W*(t) curve. The potentiostatic activation was stopped when the MEA current reached the stationary value. The characteristic time of potentiostatic activation was ~1000–1200 s (see Figure 4). The current–voltage characteristics of the MEAs were further investigated using Elins P-40X potentiostat as an electronic load.

## 3. Results and Discussion

The impedance spectra of the co-PNIS_70/30_ membrane measured at room temperature (~300 K) and 100% humidity for in-plane and through-plane directions are shown in Figure 5. 

The in-plane impedance spectrum reveals a complex character and consists of several overlapping semicircles and practically straight-line part. The bulk polymer resistance R value was determined from impedance spectrum as intersection with the real Z’ axis (arrow on Figure 5a). The proton in-plane conductivity calculated using Equation (2) consist of ~96 mS/cm. The through-plane impedance spectrum demonstrates in the Z’–Z” plane a typical dependence for the ionic conductor measurements. The film resistance R was determined as shown on Figure 5b, and the proper conductivity is equal to 60 mS/cm according to Equation (3).

The anisotropy of the conductivity can be attributed to the formation of a more hydrated layer on the polymer film surface during its exposure to humid atmosphere. However, for applications, only the through-plane conductivity is important. The through-plane conductivity of co-PNIS_70/30_ membrane turns out to be similar with that of Nafion^®^ 212 (~60 mS/cm) under the same environmental conditions. Thus, the co-PNIS membrane shows a comparable proton conductivity as Nafion^®^, rendering it a promising competitor for fuel cell applications.

The self-diffusion coefficient of water is another important parameter, which characterizes the fuel cell operation at high current densities [26,27,28,29]. Figure 6 shows the temperature dependence of *D_NMR_* in the co-PNIS_70/30_ membrane hydrated at 100% humidity and measured in the range from 213 to 353 K. We see that two temperature ranges with different Arrhenius behaviors of *D_NMR_*(T) can be distinguished. Specifically, activation energies of 0.24 and 0.33 eV are observed above and below a crossover at T = 250 K, respectively. 

It is important to note that water remains mobile down to very low temperatures of ~200 K, and the observed change in the *D_NMR_*(T) dependence may be associated with the water vitrification processes [26]. This *D_NMR_*(T) behavior is typical for various perfluorinated sulfopolymers [30,31,32]. For comparison, the self-diffusion coefficient of the Nafion^®^ 212 membrane measured under identical environmental conditions [30] shows qualitatively similar *D_NMR_*(T) dependence. The crossover in *D_NMR_*(T) occurs at approximately the same temperature (T_c_ = 253 for co-PNIS_70/30_ and T_c_ = 255 K for Nafion^®^ 212) and the activation energies coincide within the experimental error for both polymers at high (T > T_c_) and low (T < T_c_) temperatures. However, the diffusion coefficients *D_NMR_*(T) are a factor of ~3 lower for the co-PNIS_70/30_ membrane than for Nafion^®^ at all studied temperatures. In Nafion^®^, water diffuses through a system of nanochannels, while, in co-PNIS membranes, it diffuses through a three-dimensional network of hydrogen bonds, which breaks into fragments with a characteristic size of ~5 nm (the so-called confined water state) [33]. It can be concluded that, at high water content, the hydrogen-bond network in the co-PNIS membrane is somewhat less suited for fast water migration than the channel-like structure of Nafion^®^.

The current–voltage (*I-U*) and current–power (*I-W*) characteristics of MEAs based on the co-PNIS_70/30_ membrane with attached and pressed electrodes are shown in Figure 7. It can be seen that the maximum power of the MEA with pressed electrodes amounts to ~370 mW/cm^2^, while the power does not exceed a value of ~180 mW/cm^2^ for the MEA with attached electrodes. Moreover, the *I-U* dependencies are significantly nonlinear at high current densities for both samples, while a water drainage is limited by diffusion. The nonlinearity of the current–voltage characteristic begins at current densities around 350 mA/cm^2^ for the MEA with attaching electrodes. The reason can be a poor contact at the membrane–electrode interface. The *I-U* non-linear behavior of the MEA with pressurizing electrodes is observed at significantly higher current densities around 800 mA/cm^2^ and can be caused by a flooding of the cathode during water generation by means of chemical reaction and electro-osmotic drag [27,28,29]. The water supply to the cathode is dominated by electro-osmotic drag [27,28,29]. A water concentration gradient occurs in the membrane and electrodes during the fuel cell operation, and the reverse water outflow from the cathode is realized by means of diffusion processes. If both flows (electro-osmotic and reverse diffusion) are similar in magnitude, cathode flooding will not occur. Figure 6 shows that the self-diffusion coefficient for co-PNIS_70/30_ membrane at 100% humidity and all temperatures are around three times lower than for Nafion^®^. This can be the limiting factor when striving for MEA with high current densities. Notwithstanding, the maximum output power of 370 mW/cm^2^ for the MEA with pressing electrodes is quite acceptable for industrial applications, in particular when considering that, compared with Nafion^®^, the fabrication of the co-PNIS membranes is much cheaper and necessitates fewer precautions. However, final conclusions about the usability require additional investigations of the performance of fuel cells with this membrane during operation, including MEA degradation studies using various protocols [34,35].

## 4. Conclusions

This work describes a method for the synthesis of sulfonated polynaphtholenimide co-PNIS_70/30_ polymers with a high molecular weight. Membrane films were prepared by casting a polymer solution and studied by impedance spectroscopy and SFG NMR diffusion. It was found from impedance spectroscopy measurements that proton conductivity of the polymer films possesses some anisotropy and is equal to ~96 mS/cm in-plane and ~60 mS/cm through-plane directions. SFG NMR diffusometry showed that, in the temperature range from 213 to 353 K, the ^1^H self-diffusion coefficient of the co-PNIS_70/30_ membrane behaves similar to Nafion^®^ at the same humidity and is about one third of Nafion^®^ NR212 diffusion coefficient. The optimal MEAs with co-PNIS_70/30_ membranes was characterized by maximum output power of ~370 mW/cm^2^ at 80 °C. It allows considering sulfonated co-PNIS70/30 polynaphthoyleneimides polymer as a prospective proton-exchange membrane for hydrogen–air fuel cell.

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
