# Peer review of "Preparation and Study of Sulfonated Co-Polynaphthoyleneimide Proton-Exchange Membrane for a H2/Air Fuel Cell"

_materials, 2020, doi:10.3390/ma13225297_

Round 1
Reviewer 1 Report
This manuscript will be interesting for the readers of Materials, it covers the scope of the Journal. Even though the study is limited, (i.e. formulation of the sulfonated co-polynaphtoyleneimide membrane and studying its fundamental properties for low-temperature fuel cell application), that has been covered well. But a more than a major revision is required before accepting the manuscript for publishing in Materials.
Here are a few commends for improvement,
- While the membrane formulation process is not exceeded 140 °C, why the post-curing temperature is exceeding to 150 °C (for 2 days) after removing from the glass.
- In Eqn 2, the parameter “d”should be a constant that depends on the cell configuration. So it is better to mention its exact value. Similarly, the term “S” (pt area) in Eqn 3 too.
- It is mentioned that used, samples with different thicknesses for through-plane and different distances between the contacts for in-plane conductivity measurements. But just single conductivity values are given in the result and discussion for each measurement. Suggest adding a table to show the conductivity values at above mentioned different parameters.
- What are the sample thicknesses (through-plane) and distances between the contacts (in-plane) for the measurements that are shown in Figure 5? Since these parameters are essential to reproduce the reported conductivity.
- It is mentioned that additional investigations are required to understand the membrane's performance in fuel cell operation. Suggest adding a brief note from similar literature findings.
- The conclusion section is missing in the manuscript.
Author Response
Dear reviewer,
Thank you for your definite and valuable remarks, according which we make the following changes in the text:
While the membrane formulation process is not exceeded 140 °C, why the post-curing temperature is exceeding to 150 °C (for 2 days) after removing from the glass.
The duration of vacuum treatment at 150C is corrected.
In Eqn 2, the parameter “d”should be a constant that depends on the cell configuration. So it is better to mention its exact value. Similarly, the term “S” (pt area) in Eqn 3 too.
The exact geometrical values are given in the revised version of the manuscript.
It is mentioned that used, samples with different thicknesses for through-plane and different distances between the contacts for in-plane conductivity measurements. But just single conductivity values are given in the result and discussion for each measurement. Suggest adding a table to show the conductivity values at above mentioned different parameters.
We completely agree with the remark and add the corresponding table to text.
What are the sample thicknesses (through-plane) and distances between the contacts (in-plane) for the measurements that are shown in Figure 5? Since these parameters are essential to reproduce the reported conductivity.
The geometrical data is added on caption to the Figure 3.
It is mentioned that additional investigations are required to understand the membrane's performance in fuel cell operation. Suggest adding a brief note from similar literature findings.
We add corresponding reference which contain detail information on protocols.
The conclusion section is missing in the manuscript.
The conclusion is added.
All changes in the revised version of the manuscript are given by the red color.
Reviewer 2 Report
The authors report on the preparation of a co-PNIS membrane for MEA in PMEFC. The authors explain why replacing standard Nafion is of interest and compare their results with a Nafion membrane. The methods and the discussion are very well put together and the manuscript is easy to read and understand. I have a few comments:
Eq.1 - Maybe this has happened due to a PDF error, but the resolution of the equation could be improved;
Figure 3 - Can formatting of a) and b) be made the same? (e.g. legend, axis, size)
Results and Discussion, 2nd paragraph - the authors state that only the through-plane conductivity is important for applications. It would be important then for the authors to also explain here why the in-plane measurements were made at all.
I was very surprised to find there is a complete lack of a Conclusions section which one would expect to see. I am forced to recommend a Major Revision because of this.
I would also suggest that the article title is revised as it is not clear and not consistent with the formatting or grammar used throughout the manuscript.
Author Response
Dear reviewer,
Thank you for your definite and valuable remarks, according which we make the following changes in the text:
Eq.1 - Maybe this has happened due to a PDF error, but the resolution of the equation could be improved;
We improve resolution of the Eq. 1
Figure 3 - Can formatting of a) and b) be made the same? (e.g. legend, axis, size)
The Fig.3 is redid.
Results and Discussion, 2nd paragraph - the authors state that only the through-plane conductivity is important for applications. It would be important then for the authors to also explain here why the in-plane measurements were made at all.
We add our motivation on the conductivity anisotropy measurements.
I was very surprised to find there is a complete lack of a Conclusions section which one would expect to see. I am forced to recommend a Major Revision because of this.
The conclusion was lost during text preparations. The mistake is fixed.
I would also suggest that the article title is revised as it is not clear and not consistent with the formatting or grammar used throughout the manuscript.
We agree with reviewer and the title is changed.
All changes in the revised version of the manuscript are given by the red color.
Round 2
Reviewer 1 Report
The authors had responded to the comments appropriately, and the modifications are satisfactory. So recommended this manuscript for publishing in Materials in its present form.
Reviewer 2 Report
The authors addressed all my comments and the paper is suitable for publication.